

# Organic eutrophication increases resistance of the pulsating soft coral *Xenia umbellata* to warming

Svea Vollstedt[1], Nan Xiang[1,2], Susana Marcela Simancas-Giraldo[1] and Christian Wild[1]

[1] Faculty of Biology and Chemistry, Universität Bremen, Bremen, Germany
[2] WG Tropical Marine Microbiology, Leibniz Centre for Tropical Marine Research, Bremen, Germany

## ABSTRACT

Recent research indicates that hard corals in a process that is termed phase shift are often replaced by soft corals in reefs. The simultaneous occurrence of local (i.e. organic eutrophication as highly under-investigated parameter) and global (i.e. ocean warming) factors may facilitate these phase shifts as hard corals are negatively affected by both ocean warming and organic eutrophication. Knowledge about soft coral responses to environmental change remains incomplete, although these organisms are becoming important players in reefs. The present study thus investigated the individual and combined effects of organic eutrophication (as glucose addition) and warming on the ecological data of the pulsating soft coral *Xenia umbellata*. We assessed health status, growth and pulsation rates of soft corals in a 45 day aquarium experiment, with first manipulation of organic eutrophication (no, low, medium and high glucose addition) over 21 days followed by step-wise increases in water temperature from 26 to 32 °C over 24 days. Findings revealed that glucose addition did not affect health status, growth and pulsation rates of the investigated soft corals. Under simulated ocean warming, soft corals that had experienced organic eutrophication before, maintained significantly higher pulsation rates (averaging 22 beats per minute—bpm) and no mortality compared to the controls that showed a decrease of 56% (averaging 15 bpm) in pulsation rates and mortality of 30% at water temperatures of 32 °C compared to 26 °C. This apparently positive effect of organic eutrophication on the ecological data of soft corals under an ocean warming scenario decreased with increasing water temperature. This study thus indicates that (a) organic eutrophication as additional energy source up to a certain threshold may increase the resistance of soft corals to ocean warming and (b) pulsation rates of soft corals may be used as inexpensive, easily detectable, and non-invasive early warning indicator for ocean warming effects on benthic reef communities. When comparing findings of this study for soft corals with previous results for hard corals, it can be assumed that soft corals under the predicted increases of organic eutrophication and warming gain more and more competitive advantages. This may further facilitate phase shifts from hard to soft corals in warming reefs.

Corresponding author
Svea Vollstedt,
sv_vo@uni-bremen.de

# INTRODUCTION

Coral reefs are biodiversity hotspots and among the most productive ecosystems in the world (*Connell, 1978*). Although they cover only less than 1% of the ocean floor, coral reefs are of high economic value (*Moberg & Folke, 1999*). Besides their ecological and economic importance, coral reefs are extremely vulnerable ecosystems (*Connell, 1978*; *Hoegh-Guldberg, 1999*; *Moberg & Folke, 1999*). During the last decades, an alarming change with a sharp decline in species diversity and habitats has been described for coral reefs (*Sebens, 1994*). Approximately 20% of hard corals worldwide are already irreversibly destroyed and an additional 30% are strongly damaged (*Bange et al., 2017*). There is hardly any unaffected reef left (*Hughes et al., 2003*; *Jackson et al., 2001*).

Hard corals produce a rigid aragonite endoskeleton and create a unique and stable inorganic carbonate substrate and 3D framework system (*Scheffers et al., 2003*). Therefore, reef-building corals are essential ecosystem engineer organisms and highly important for reef-associated biodiversity (*Wild et al., 2011*; *Wild & Naumann, 2013*). Recently, several studies suggest hard corals respond negatively to ocean warming and/or elevated nutrient concentrations (*Fabricius et al., 2013*; *Wooldridge & Done, 2009*). At many locations in the Indo-Pacific phase shifts from hard to soft corals have been observed (*Norström et al., 2009*). Soft corals therefore may become essential to reef ecosystems in the future (*Hoegh-Guldberg, 1999*).

Coral reefs worldwide are susceptible to the effects of global warming. Over the last 50 years, the sea surface temperature has raised by 0.11 °C per decade and it is estimated to continue warming by 0.6–2.0 °C before 2100 (*Chaidez et al., 2017*). Another global threat is the rising atmospheric carbon dioxide $CO_2$ it reduces the ocean pH and causes ocean acidification (*Doney et al., 2009*; *Inoue et al., 2013*; *Rodolfo-Metalpa et al., 2010*).

Organic eutrophication is a highly under-investigated factor that may contribute to coral reef decline (*Rabalais et al., 2009*). Terrestrial runoff washes nutrients, sediments and pollutants from urbanised or fertilised seaside into coastal waters and thereby represents a major sources of organic eutrophication (*Fabricius et al., 2013*). Furthermore, domestic and industrial contaminants increase the degree of organic contamination (*Braga et al., 2000*). The excess input of dissolved organic carbon (DOC) is a typical organic eutrophication in tropical waters. Nevertheless, DOC is also released naturally by algae, seagrass and corals (*Haas et al., 2010a*; *Haas et al., 2010b*; *Smith et al., 2006*). *Kline et al. (2006)* reported that DOC increased coral mortality due to a disruption of the symbiosis between the hard coral *Orbicella annularis* (Ellis & Solander, 1786) and its associated microbiota. Coral reef ecosystems depend on the symbiosis between coral hosts and their intracellular photosynthetic dinoflagellates family *Symbiodiniaceae* (see *LaJeunesse et al., 2018*). Nevertheless, the associated bacteria importantly contribute to the functioning of the entire coral. Nitrogen (N) cycling microbes are commonly

associated with corals and provide N to both, the *zooxanthellae* and the coral host (*Pogoreutz et al., 2017*; *Rädecker et al., 2015*). Since organic eutrophication provides fast and directly digestible sugar, microbes enhance their growth (*Kuntz et al., 2005*; *Pogoreutz et al., 2017*). The proliferation and thereby higher microbial activity is detrimental because of the enhanced O$_2$ depletion which could have severe consequences for ecosystem functioning and the coral metabolism (*Wild et al., 2010*). Although temporal hypoxia is common in reef ecosystems, severe hypoxic events can cause widely coral mortality (*Simpson, Cary & Masini, 1993*). Different studies suggest, that high temperature stimulates microbial proliferation and in addition increase the nitrogenase activity (*Cardini et al., 2016*; *Santos et al., 2014*). This excess (microbial) fixed nitrogen may lead to a disruption of the N limitation of hard coral associated zooxanthellae, subsequently leading to coral bleaching via a cascade of processes described by *Pogoreutz et al. (2017)*. Such bleaching susceptibility of corals may be species-specific (*Hoegh-Guldberg, 1999*), and knowledge about soft coral bleaching responses is scarce.

Soft like hard corals depend on the metabolic communication between coral host, endosymbiotic zooxanthellae and a diverse microbial community. *Xenia umbellata* (Lamarck, 1816) is a common soft coral mainly distributed in the Indo-Pacific and the Red Sea and occurs there in high abundances (*Al-Sofyani & Niaz, 2007*; *Janes, 2014*). This soft coral colonises both sand slopes and hard substrates in a water depth of 3–25 m (*Janes, 2014*). The distinctive, rhythmic pulsation of their tentacles, first recorded by Lamarck about 200 years ago, makes them remarkable (*Kremien et al., 2013*). This mechanism may enhance the photosynthetic efficiency and prevent refiltration as *Kremien et al. (2013)* detected for the closely related species *Heteroxenia fuscescens*. The enhancement of photosynthesis may be maintained by fast removal of excess oxygen at the coral surface and may account for the reduced gastrovascular capability of pulsating xeniid corals (*Kremien et al., 2013*). In addition the tentacles support the ability to be a heterotrophic suspension feeder (*Al-Sofyani & Niaz, 2007*). However *X. umbellata* is able to live autotrophic based on photosynthesis products (*Schlichter, Svoboda & Kremer, 1983*) from the zooxanthellae.

There are pronounced gaps of knowledge in relation to the response of soft corals to a combination of local and global stressors since most previous research concentrated on hard corals and the effect of either a local or a global stressor. In addition, organic eutrophication is a highly under-investigated stress factor, although it commonly occurs in coastal areas. This study thus aims to answer the following research questions: (1) How does organic eutrophication (as glucose addition) affect pulsation and growth rates along with the health of the soft coral *X. umbellata*? (2) Does organic eutrophication (as glucose addition) influence the resistance of the soft coral *X. umbellata* to ocean warming? We aimed to answer these research questions by implementing a laboratory manipulation experiment over 45 days. With the resulting experimental data and the available literature, we compare the responses of soft in comparison to hard corals to organic eutrophication and warming in order to assess potential winners and losers under the predicted future scenarios.

## MATERIALS AND METHODS

### Experimental setup

The soft corals used in the experiments are originally from the northern Red Sea but kept in the maintenance aquarium of Marine Ecology Department since more than 2 years prior to the experiments. The soft coral genus *Xenia* is very common in the Indo-Pacific reef ecosystems (*Al-Sofyani & Niaz, 2007*; *Janes, 2014*). We identified the soft coral according to *Reinicke (1997)*, as species *X. umbellata* Lamarck, 1816, which is a common pulsating *Xenia* species from the Red Sea. Five big colonies (5 × 7 × 12 cm) were fragmented into 160 small colonies (1–2 cm in width) using a scalpel. Subsequently, these colonies were attached to cubical-shaped calcium carbonate coral holders (1 × 1 cm) with rubber bands. The colonies could heal and grow on the holders for 7 days. Thereafter, the rubber bands were cut off and the colonies were evenly distributed among 12 experimental aquaria tanks (volume: 60 L; details please see below) and left there for acclimation for 10 more days. The tanks were still connected with the main tank and a stable water flow was maintained. After acclimatisation, the first ecological data were recorded as a baseline (day 0) for the experiment.

The experiment was conducted using an aquarium tower which was made out of 12 individual tanks. The tanks were arranged in three levels with four tanks in one level. Each tank had a volume of 60 L, consisting of technical and experimental parts. The volume of water in the tanks was 50 L. Each tank was equipped with 20 *X. umbellata* colonies. In the technical part of the tanks a thermostat (3613 aquarium heater. 75 W 220–240 V; EHEIM GmbH and Co. KG, Germany) and a pump (EHEIM CompactOn 300 pump; EHEIM GmbH and Co. KG, Germany) were installed. To maintain a stable temperature and prevent the water from cooling, styrofoam boards were placed on top of the aquaria when needed. The LED light (Royal Blue—matrix module and Ultra Blue White 1:1—matrix module, WALTRON daytime® LED light, Germany) was a source of heat, so it was placed on top of the aquaria and adjusted in height. Light intensity of 120.8 ± 10.2 µmol quanta $m^{-2}$ $s^{-1}$ and a day-night rhythm of 12–12 h.

Further, 10% of the water was replaced on a daily basis, assuring close to natural sea water parameters with a high renewal rate of seawater in the tanks. Maintenance conditions were kept constant and the algae growth was kept within limits by repeated cleaning. Water parameters were monitored twice a week. Temperature and salinity were measured every day and adjusted manually. In the first part of the experiment the temperature was kept stable around 26 °C, while in the second part an increase of temperature was performed. There were no significant differences between treatments in salinity (average: 35.4 ± 0.4). Chemical parameters for all tanks are summarised in Table 1.

### Manipulation of organic eutrophication and temperature

In nine aquaria, organic eutrophication was manipulated by daily additions of D-Glucose anhydrous (purity: 99%, Fisher Scientific U.K. Limited, Loughborough, UK). To maintain three different treatments, we measured TOC as a proxy for glucose addition and calculated the addition to reach: 10 mg/L as low concentration, 20 mg/L as medium

**Table 1 Mean values (±SD) of water parameters maintained in all tanks.**

| Parameter | Mean values (±SD) |
| --- | --- |
| pH | 8.1 ± 0.1 |
| KH | 7.2 ± 1.0 |
| $Ca^{2+}$ | 447.8 ± 66.4 ppm |
| Mg | 1,493.1 ± 293.5 ppm |
| $NO^{2-}$ | <0.01 ppm |
| $NO_3$ | <0.5 ppm |
| $NH_4$ | <0.05 ppm |
| $PO_4$ | <0.02 ppm |

concentration and 40 mg/L as high concentration. After the first addition, we monitored the glucose addition regularly with a Total Organic Carbon Analyzer TOC-L (Shimadzu Corporation, Kyoto, Japan) in order to adjust glucose addition as response to consumption. Every afternoon after sampling, the missing amount of glucose was added in form of a stock solution (D-Glucose, concentration 40 mg/mL) to achieve the desired final concentrations (for more details see supplied data). This procedure was continued throughout the experiment, while the stock solution was prepared daily. The other three tanks were maintained at ambient glucose levels (control 2–3 mg/L). The glucose concentrations were based on previous studies that manipulated organic eutrophication (*Kline et al., 2006*; *Pogoreutz et al., 2017*).

The temperature was kept stable at 26 °C for the first three weeks of the experiment. After day 21, the temperature was increased by 2 to 28 °C within 5 days and then kept stable for 3 days. This was repeated two more times until the maximum temperature of 32 °C was reached. The light intensity was measured with a LI-1400 Data Logger (LI-COR, Inc., Lincoln, NE, USA) once a week over the entire duration of the experiment. This resulted in a mean light intensity of 120.8 ± 10.2 µmol quanta $m^{-2}$ $s^{-1}$. The temperature treatment was chosen to imitate the possible natural temperature in the Red Sea, a distribution area of *X. umbellata*. In the southern Red Sea, a temperature of 33 °C is reached between late July and mid-August. Here the organisms may be yet close to their thermal limits (*Chaidez et al., 2017*). Thereby, *Strychar et al. (2005)* used 32 °C as a maximum temperature as well and found that *Xenia* spp. is susceptible to relatively small temperature changes.

### Ecological assessments

We used the pulsation and growth rates to assess the sublethal effects. Pulsation rates were always quantified by the same observer and the same methodology. Samples were taken 13 times in the first 3 weeks having a higher resolution in the first 2 weeks and fewer counting in the last. During the second part of the experiment pulsation rates were counted before and after each temperature increase. Three random colonies from each tank were selected about noon every sampling day to avoid the corals being disturbed via glucose addition or other measurements. We decided to use pulsation per minute as a comparable

unit. Pulsation rates that occurred in 1 min were then counted for a single polyp by using a stopwatch. A complete pulsation was defined as the time that a polyp needed to open and close all its tentacles. After 7 days the pulsation was counted 30 s to save time, but was calculated to pulsation rates per minute afterwards.

To calculate the growth rate (new polyps per day), always the polyps of the same three colonies per tank were counted over time. Therefore, the colony was taken out of the tank without air exposure, and the number of polyps of every colony was counted using tweezers in order to avoid double-counting. With the data of polyp numbers, the growth rates of new polyps per day were determined.

## Data analyses

Statistical analyses were performed using GraphPad Prism 8 and SPSS 17.0. All data presented in the text, figures and tables are expressed as means ± SD. The significant differences in variables between treatments were analysed by one-way analysis of variance (ANOVA) with Duncan's test. For each timepoint and temperature an individual ANOVA was performed. The assumptions for an ANOVA were checked a priori. The effect $p < 0.05$ was regarded as significant and $p < 0.01$ was considered as highly significant.

# RESULTS

## Glucose tank concentration monitoring

The mean TOC concentrations were measured over the first and last week of the experiment 18 h after the prior glucose addition. Our mean values were $2.95 \pm 0.36$ mg L$^{-1}$ (control), $5.84 \pm 0.99$ mg L$^{-1}$ (low), $11.72 \pm 2.09$ mg L$^{-1}$ (medium), $26.57 \pm 4.56$ mg L$^{-1}$, respectively.

## Mortality

After 3 weeks exposure to elevated organic nutrient concentrations, no mortality was detected. The mortality of corals was observed during the last days of the temperature experiment. 30% of the control corals with increased temperature died. Remaining colonies were shrunken and showed visible signs of stress. In comparison, no coral of the glucose treatments died until the end of the experiment and remaining colonies looked less damaged.

## Pulsation rates

On baseline day 0, all pulsation rates ranged from 35 to 44 beats min$^{-1}$. After glucose addition for 21 days, pulsation rates showed no significant change among all groups ($p = 0.198$) (Fig. 1A).

With the first temperature increase to 28 °C, there was no significant difference between any treatment ($p = 0.153$; Fig. 1B) and also no difference compared to control colonies at 26 °C. However, the increased temperature control showed a significant decrease in pulsation rate compared to 26 °C and a significant decrease ($p < 0.05$) by 52 and 33% compared to all glucose addition treatments at 30 and 32 °C, respectively. Soft corals in the

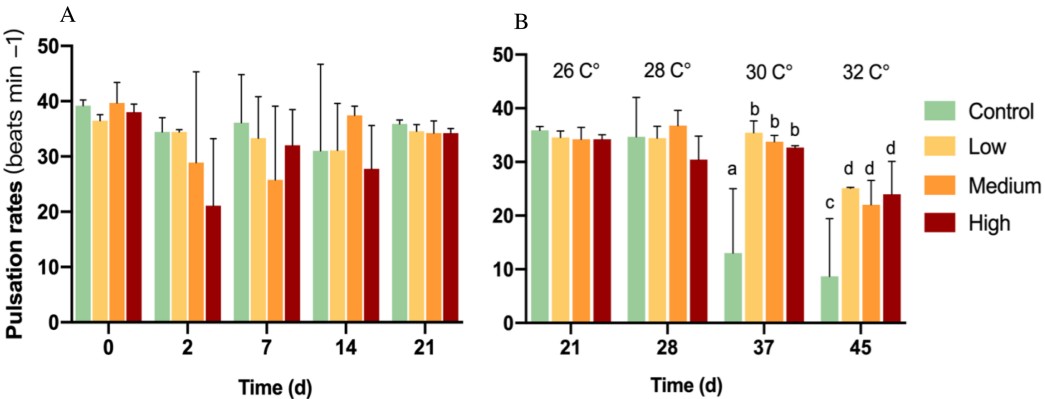

**Figure 1 Pulsation rates of corals at different glucose concentrations during the experiment.** Columns indicate mean values of three replicates with error bars providing the respective SD. (A) Exclusive glucose experiment over time. (B) Glucose treatments with step-wise increased temperature over time. (B) starts with day 21 at 26 °C as a starting point for the temperature increase. Letters indicate statistically significant differences ($p < 0.05$).

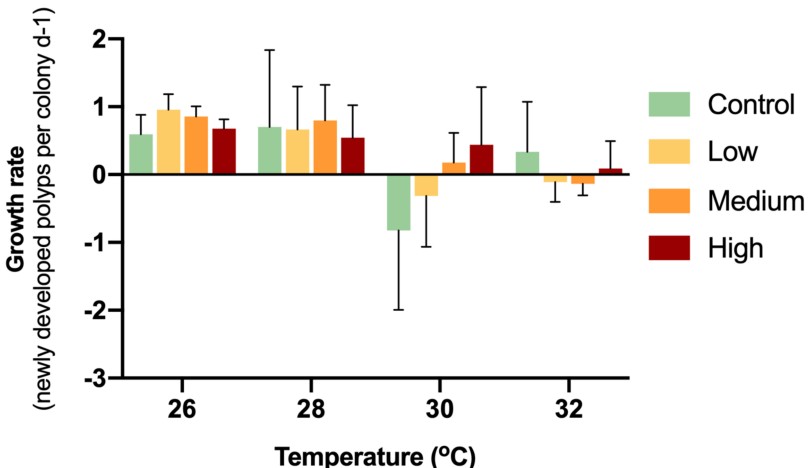

**Figure 2 Growth rates of corals with different glucose concentrations and temperatures during the experiment.** Columns indicate mean values of three replicates with error bars providing the respective SD. The growth rates did not show any statistically significant differences ($p > 0.05$).

glucose addition treatments decreased their pulsation by around 30% at 32 °C compared to 26 °C, but they exhibited over two-times higher pulsation rates than the control in 32 °C.

## Growth rates

While the control colonies exhibited a growth rate of 0.59 new polyps d$^{-1}$, the enriched treatments caused growth rates of 0.96 (low), 0.86 (medium) and 0.68 (high) new polyps d$^{-1}$ (Fig. 2). Thereby, glucose enrichment maintained the soft coral growth rate by 62% (low), 45% (medium) and 13% (high) compared to the controls.

During the first week of temperature increase to 28 °C, a positive growth rate was detected in all treatments. With increased temperature to 30 °C, medium and high glucose treatments increased in polyp numbers, while the other treatments showed negative

growth rates. With increased temperature to 32 °C, only the high glucose and ambient glucose group showed positive growth rates. It is worthy to note, that in the high glucose treatment, corals continued to slowly grow even until the end of the experiment. There were no significant differences ($p = 0.09–0.70$) between treatments at particular temperatures.

## DISCUSSION

### How does organic eutrophication (as glucose addition) affect pulsation and growth rates along with the health of the soft coral *X. umbellata*?

Findings revealed that glucose addition did not affect polyp growth and pulsation rates. The pulsation rates of *X. umbellata* showed a slight decrease in all treatments after 21 days of glucose exposure, although pulsation rates ranged within the limit of in situ pulsation rates of 34 to 42 beats $min^{-1}$ (*Kremien et al., 2013*). Our findings do thus not suggest a disturbance of the soft coral caused by glucose addition. The comparison of all treatments at ambient temperature showed positive and similar growth rates of *X. umbellata*. Other experiments showed hard coral mortality with comparable glucose loading of 25 mg/L over a period of 30 days (*Kline et al., 2006*; *Kuntz et al., 2005*). *Kline et al. (2006)* proposed a species-specific response to glucose loadings likely mediated by dynamics of the symbiont community. In *X. umbellata*, the positive response could point towards a symbiont community that can benefit from moderate glucose addition as energy source. Furthermore, soft corals are known to have antimicrobial properties in their coral mucus (*Ritchie, 2006*) and may have lower growth of potentially pathogenic bacteria compared to hard corals. In general, the disruption of the functioning of the symbiosis can lead to lower or stagnating growth rates up to mortality (*Hoegh-Guldberg, 1999*). Our results show less detrimental effects on soft corals under these specific experimental conditions. The findings are contrary to previous findings when sugar-stressed hard corals exhibited negative changes in photosynthetic and nutrient cycling properties which are very similar to bleaching responses (summarized in Table 2) (*Pogoreutz et al., 2017*).

Also, the heterotrophic feeding ability may be of potential importance for *X. umbellata* when coping with eutrophication. It is an additional energy source, most likely enhanced by polyp pulsation, and it may maintain the host functions even with autotrophic disruption (*Wild & Naumann, 2013*; *Wooldridge, 2013*). Switching from photo-autotrophy to mixotrophy, could have enhanced the growth rates of *X. umbellata* in treatments with elevated glucose levels. It needs to be investigated to what extent a switch to heterotrophic feeding is possible, because previous findings are contradicting. While *Gohar (1940)* and *Schlichter, Svoboda & Kremer (1983)* classified *Xenia* as completely autotrophic, *Fabricius & Klumpp (1995)* proposed, that *Xenia* is dependent on both feeding strategies. Soft corals have poor predatory abilities compared to hard corals (*Fabricius & Klumpp, 1995*) but this does not mean they cannot feed on particulate organic matter such as phytoplankton, labile detritus, microzooplankton and bacteria

**Table 2 Enrichment experiments with different organic carbon concentrations and coral species.** All previous studies were conducted with hard corals in ambient reef temperature and showed negative impacts of elevated glucose compared to control treatments with ambient glucose levels. The present study showed neutral effects of glucose addition in combination with ambient temperature and positive effects of glucose on pulsation rates and mortality of soft corals compared to control treatments in elevated temperature. Negative (−), positive (+) and neutral (o) implications for corals are indicated.

| Study | Species | Glucose concentration (mg/L) | Time (days) | Results (compared to control) | Implications |
|---|---|---|---|---|---|
| Pogoreutz et al. (2017) | Pocillopora verrucosa | 10 | 28 | Pronounced paling | − |
| Kline et al. (2006) | Orbicella annularis (Ellis & Solander, 1786) | 12.5 25 | 30 | 5-fold higher, significant mortality in both levels | − |
| Kuntz et al. (2005) | O. annularis, Agaricia tenuifolia, Porites furcata | 5 25 (lactose addition) | 30 | Species-specific higher mortality rate | − |
| Haas, Al-Zibdah & Wild (2009) | Acropora | 110 | 90 | Reduced chlorophyll a tissue concentration, colour decrease | − |
| This study | Xenia umbellata | 10 20 40 | 21 | No significant effect | o |
| This study | Xenia umbellata | 10 20 40 | 45 | Significantly higher pulsation rate, reduced mortality in all treatments | + |

(*Fabricius & Dommisse, 2000*). Furthermore, glucose addition can have direct effects being an energy source for reef associated microbes or indirect effects as increased turbidity. Especially within closed systems, indirect effects have a considerable influence. In our experiment, the indirect effect of increased turbidity was particularly strong to observe (no measurements were performed) in the high glucose treatment during the last week of the experiment. This could have reduced the ability of soft corals to utilise energy via photosynthesis. High turbidity may have decreased the possible combined effect of photo-autotrophically and heterotrophically fixed carbon energy sources.

### Does organic eutrophication (as glucose addition) influence the resistance of the soft coral *X. umbellata* to ocean warming?

Under simulated ocean warming, soft corals that had experienced organic eutrophication before, maintained significantly higher pulsation rates and showed no mortality compared to the controls. The reduction of pulsation could be the first step of *X. umbellata* to cope with temperature stress as energetic investment is minimised. Therefore, pulsation rates may be used as inexpensive, easily detectable and non-invasive early warning indicator for ocean warming effects on benthic communities. Already 64 years ago, *Horridge (1956)* published work about "the responses of *Heteroxenia* to stimulation and to some inorganic ions". The only other related studies focused on the toxicity and sublethal effects of crude oil (*Cohen, Nissenbaum & Eisler, 1977*; *Studivan, Hatch & Mitchelmore, 2015*). *Cohen, Nissenbaum & Eisler (1977)* found a reduction of

pulsation with a decline to less than 50% when exposed to crude oil. *Studivan, Hatch & Mitchelmore (2015)* suggested *Xenia elongata* as bioindicator for coral species in other locations than the Indo-Pacific reefs because of its high sensitivity to changes in water quality.

Nevertheless, pulsation provides great benefits for the coral colony. The enhancement of photosynthesis during the day and the prevention of refiltration during both day and night (*Kremien et al., 2013*). Thus, a maintained pulsation may protect *X. umbellata* exposed to warming. In treatments with elevated glucose levels, the switches from primarily photo-autotrophically to combining heterotrophically fixed carbon and photosynthesis, could have supported energetic investment required for pulsation. Organic eutrophication as additional energy source may thus increase the resistance of soft corals to ocean warming.

In general, heterotrophic feeding can support corals when photosynthetic activity is reduced by photodamage (*Borell & Bischof, 2008*; *Wooldridge, 2013*). In limited amounts, glucose addition may enhance the availability of photosynthetically fixed carbon to the coral host. This excess fixed carbon may be stored in tissues as lipids and create an important energy reserve or directly be channelled into growth and reproduction (*Wild & Naumann, 2013*; *Wooldridge, 2013*). With low and moderate levels of autotrophic disruptions, the coral host may also be able to consume the existing energy reserves (*Grottoli, Rodrigues & Palardy, 2006*) that likely have been fuelled by previous glucose addition.

Throughout the temperature experiment, little mortality was noted among soft coral colonies, but they did not maintain their normal appearance and showed visible signs of stress. While the increased temperature up to 28 °C showed a positive growth rate in all treatments (not notably different from growth rates calculated at 26 °C), in general the growth rates decreased with increasing temperature.

However, an increase up to 28 °C did not negatively affect the soft corals. In the Red Sea, an among others distribution area of *X. umbellata*, 28 °C are common water temperatures (*Chaidez et al., 2017*). Therefore, the colonies may potentially be resistant to a short exposure time. 30 °C showed the first evidence of negative influence of elevated temperature on *X. umbellata* colonies. Other studies concluded the same and showed higher rates of mortality with temperatures >30.5 °C (*Cantin et al., 2010*; *Strychar et al., 2005*). Some studies even predict xeniids to be more susceptible to temperature stress than other Octocorallia (*Sammarco & Strychar, 2013*; *Strychar et al., 2005*). In the present study, no significant difference in growth rates could be detected between corals at different glucose treatments. In addition, the high glucose treatment continued to slightly grow even until the end of the experiment, on the contrary to the lowest glucose concentration. These findings may suggest that glucose addition supports *X. umbellata* to maintain growth and increase the ability to withstand the effect of elevated temperature.

In contrast to our findings, high levels of glucose caused strong negative effects on two hard corals, showing that coastal eutrophication produces an additional stress factor that even outweighed nutritional benefits (*Fabricius et al., 2013*). These results

highlight the fact that coral mortality patterns may depend on each type of stressor, the species of coral, and the duration of exposure time as *Kuntz et al. (2005)* mentioned.

## Ecological perspective

Human activity will likely further increase organic eutrophication as a local stressor of marine ecosystems, and climate change will further increase ocean warming. Even achieving the ambitious goal of 1.5 °C of global warming under Paris Agreement, will lead to a loss of 70–90% of reef building corals versus today (*Hoegh-Guldberg et al., 2018*). Given this severity of coral loss, understanding the ecophysiology of fast-growing xeniids is an important aspect for future predictions (*Wild & Naumann, 2013*). Besides, *Norström et al. (2009)* reported benthic reef community shifts which are often related with changes in the dynamics of bottom-up factors (e.g. nutrient enrichment). Xenia was observed to overgrow large fields of rubble in the Komodo National Park in Indonesia where blast fishing destroys living coral at an extensive rate (*Fox et al., 2003*).

Different recovery and reproduction strategies (r- and k-strategies) are also involved in the abundance of coral types after severe mortality or bleaching events with increased frequency (*Hoegh-Guldberg, 1999*). Soft corals often have high fecundity and several dispersal modes to rapidly colonise damaged reef areas (*Fox et al., 2003*), while hard corals may not be able to get mature or reproductive before the next period of intensive environmental stress. Soft corals have a competitive advantage against hard corals because of higher resilience to temperature induced bleaching and ocean acidification (*Inoue et al., 2013*; *Wild & Naumann, 2013*). Findings of a study on *Ovabunda macrospiculata* suggest the octocoral tissue to have a possible protective role against sclerite loss under acidic conditions (*Gabay et al., 2014*). In addition powerful chemical defence mechanisms benefit soft corals (*Fox et al., 2003*).

The results of the present study suggest that *X. umbellata* has a high resistance to a combination of organic eutrophication and warming. These findings show contradictory results to previous studies (summarised in Table 2) and speak for the fact, that soft corals respond differently than hard corals when exposed to organic eutrophication. When comparing findings of this study for soft corals with previous results for hard corals (*Haas, Al-Zibdah & Wild, 2009*; *Pogoreutz et al., 2017*), it can be assumed that soft corals under the predicted increases of organic eutrophication and warming gain more and more competitive advantages. This may further facilitate phase shifts from hard to soft corals in warming reefs.

Reef-associated biodiversity is under threat (*Sebens, 1994*). With a loss of diversity and a dominance of soft corals, a decline of important ecosystem functions is predicted. A loss of reef-building corals will lead to a decrease in 3D framework systems and lessened release of mucus (*Wild & Naumann, 2013*). Still the overall productivity of the coral reef may not necessarily decrease because of the higher photosynthesis to respiration ratio that *Kremien et al. (2013)* discovered for the pulsating *H. fuscescens* compared with non- pulsating hard and soft corals. But xeniid soft corals do not release extensive amounts of organic matter into the reef environment and therefore cannot provide to function as an energy carrier and particle trap (*Bednarz et al., 2012*; *Wild & Naumann, 2013*).

However, in this way the high economic and social capacity of coral reefs will not be preserved (*Hoegh-Guldberg et al., 2018*). Although climate change is a global issue, reducing local stressors can help in maintaining reefs and enhancing reef resilience including limiting the long-term damage (*Hughes et al., 2003*). Reducing organic eutrophication is one promising management strategy to avoid phase shifts from hard to soft corals.

## ACKNOWLEDGEMENTS

We would like to thank the Marine Ecology Group, namely Edoardo Zelli, Nadim Katzer, Meghan Kennedy, Rassil Nafeh and Claudia Dessi, at University of Bremen for technical support in the implementation of this experiment.

### Funding

This study was supported by baseline funding of University of Bremen to Christian Wild. The funders had no role in study design, data collection and analysis, decision to publish, or preparation of the manuscript.

### Grant Disclosures

The following grant information was disclosed by the authors:
University of Bremen to Christian Wild.

### Competing Interests

The authors declare that they have no competing interests.

### Author Contributions

- Svea Vollstedt conceived and designed the experiments, performed the experiments, analysed the data, prepared figures and/or tables, authored or reviewed drafts of the paper, and approved the final draft.
- Nan Xiang performed the experiments, analysed the data, prepared figures and/or tables, authored or reviewed drafts of the paper, and approved the final draft.
- Susana Marcela Simancas-Giraldo performed the experiments, analysed the data, authored or reviewed drafts of the paper, and approved the final draft.
- Christian Wild conceived and designed the experiments, authored or reviewed drafts of the paper, and approved the final draft.

### Data Availability

    The raw measurements are as a Supplemental File.

### Supplemental Information

Supplemental information for this article can be found online at http://dx.doi.org/10.7717/peerj.9182#supplemental-information.

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
