# Peer review of "Organic eutrophication increases resistance of the pulsating soft coral Xenia umbellata to warming"

_PeerJ, doi:10.7717/peerj.9182_

## Round 0.1 · original submission · Major Revisions

I now have reviews back from 2 experts in the field and while both agree that your study has merit and provides an interesting and useful contribution to the field, they also both recommend major revisions before the manuscript would become acceptable for publication.

In addition to many small issues raised by both referees throughout the text, the major issues that they point out are a deficiency in citing the prior literature and the lack of information regarding the focal taxa, before extrapolating to soft corals in general and further discussing differences between hard and soft corals. Likewise issues of clarity regarding specific hypotheses, growth rates and other physiological parameters such as bleaching could be improved.

Overall the reviews are detailed and provide specific guidance for improvement of the manuscript. While the list is rather extensive, and hence the request for major revisions, I do not see anything that I expect to be a sticking point in the revision of the manuscript, and I expect a suitably revised submission would be met with a favorable response from the referees.

I look forward to seeing your revised manuscript.

Reviewer 1 ·

Basic reporting

TABLES AND FIGURES

Table 2: This study should be cited as “This study” rather than “Vollstedt, 2019”. Haas et al. (2009) is missing in the reference list and checking their paper seems that they added 110 mg/L glucose and not 55mg/L (55 g was added to 500L). Importantly, this study shows a positive effect of glucose addition on pulsation rates, growth and survival of X. umbellata when combined with ocean warming only. Please specify this nuance and add a “neutral” implication for glucose addition at ambient temperature. If possible, please include the temperature used in the studies on hard corals.

Figure 1: What does the 26°C treatment in the right panel represent? Is it a mean of the left panel?

Figure 2: Please provide visual statistical results if they are significant.

Experimental design

MATERIALS AND METHODS
General comment: Please provide information about the geographical origin about the soft coral species and also about how was the species was identified?

Line 131: Check the unit or number of the light intensity. 2918.4 µmol quanta m-2 s-1 seems extremely high. With what kind of sensor was the light intensity measured?

Lines 158-160: Change to: “the light intensity measured as lx was converted to….”

Lines 160-162: You mention changes of additional 2.0°C according the IPCC report but your highest temperature delta is 6.0°C. There is something unclear here. Do you maybe refer to a seasonal temperature in a specific place? If yes, please develop.

Line 184: Please replace “whereas” by a linking word that does not express a contrast.
Line: 181-185: Indicate that for each timepoint and each temperature an individual ANOVA was performed. Were the assumptions for an ANOVA checked a priori? Please include.


RESULTS
Line 189-191: Please report the measured C concentrations in the tanks including means with standard deviations. Here you provide levels of glucose measured. Did you really measured glucose? I assume it was TOC with the Shimadzu analyser.

Fourth paragraph “Growth rates”: Would you have any pictures showing the new polyps? This is an interesting finding. However the information lines 215-216: “i.e. growth rates were declining with increasing glucose concentrations” should not be stated that way since the growth rates in the glucose addition treatment were higher than the control. Is there any statistics to be displayed here?

Line 219: “increased” is confusing. I’d rather suggest the use of “maintained”.

Line 224 and discussion: If there are no significant differences between treatments for all temperatures, is glucose addition really counterbalancing the detrimental effects of warming? Negative growth rates were measured at 30°C for the control but the control was again positive at 32°C whereas the glucose addition treatments ranged from negative to slightly positive...

Validity of the findings

ABSTRACT
Line 32: “increasing organic eutrophication” should be “increasing water temperature” as pulsation rates were not different between the 3 glucose treatments but decreased with increasing T.

INTRODUCTION
Lines 56-58: This sentence should be split into two to ensure a clear understanding. Please specify what ”important” means to you.
Lines 72: Please explain shortly why bacteria contribute to the functioning of coral holobionts.

Line 74: “microbes enhance their growth with sugar addition”. Please include why is this detrimental for corals (e.g. hypoxia, proliferation of pathogenic bacteria….)
Line 83: Please explain: A very SPECIAL soft coral…

Line 102: Rather talk about resistance than “resilience”. We usually talk about resilience when investigating the recovery to a stress.

DISCUSSION
- First part
Line 230: Replace health status with polyp growth rates.
Line 235: “during” should be “at”

Lines 234-236: If the comparison of all treatments at ambient temperature did not show statistical difference, just state that the growth rates were positive and similar. Otherwise, please clarify this sentence.

Line 236-245: The Kline 2006 study points mainly towards increased growth rate of microbes living in the corals’ surface mucopolysaccharide layer in response to sugar addition. They then suggest that coral mortality occurs due to a disruption of the balance between the coral and its associated microbiota. Soft coral mucus is known to have very high antimicrobial properties. I wonder if it could it also be that the sugar-induced growth of potentially pathogenic bacteria in the soft coral mucus is lower as compared to hard corals? And thus, sugar addition may have a less detrimental effects on soft corals under these specific experimental conditions.

Lines 238-241: It could be interesting to state what the genus of Symbiodiniaceae harbored by your corals was here. Is it possible to develop this part a bit more?
Lines 242-245: Could you please specify the direction of changes you mention here for photosynthetic and nutrient cycling properties? Cite the table 2.

Lines 249-250 and 277 should be “from photo-autotrophy to mixotrophy”? Or something similar but primarily photo-autotrophically is not correct and heavy.
Line 255: Soft corals may have poor predatory apparatus compared to hard corals but that does not mean they cannot feed on particulate organic matter such as phytoplankton, labile detritus, microzooplankton and bacteria (Fabricius and Dommisse, 2000).

Line 259: This makes sense but have you measured the turbidity? Also please replace “in” by “during the last week of the experiment”.

Line 260: Modify to “the ability of soft corals”

Line 263-264: Please specify that these sentences are stated for 26°C only. Was this decrease significant? Regarding Figure 2, increased glucose treatments 1) did not show decreased growth rates and 2) the highest glucose concentration always exhibited a positive growth rate for all temperatures, on the contrary to the lowest glucose concentration…

- Second part
Line 266: Please use another word than “resilience” here and throughout the manuscript.
Lines 268-273: Are there any other report of pulsation change under other environmental conditions? A comparison could be interesting to further discuss the choice of Xeniidae pulsation as early warning signal and to eventually extend it beyond warming temperatures.
Line 274: Please nuance this sentence. Is there no report on bleaching in pulsating soft coral species? See Styrchar et al. (2005).
Line 285: You mean “heterotrophic feeding” or “autotrophic disruptions”?
Lines 301-303: “colonies previously exposed to eutrophication” confuses the reader and made me wonder whether the exposure to glucose has stopped after the temperature increase? Please clarify.
Line 307: This is not a contradiction since hard and soft corals are very different organisms. However it is in contradiction to studies reporting decline of soft corals exposed to eutrophication on the field.
Lines 309-311: Could you please modify this sentence to ensure a clearer understanding?

- Third part
General comment: It could be maybe nice to add information of areas that have encountered phase-shift to Xeniidae, what triggered a change in the community structure and to primarily focus on Xeniidae details before talking about soft corals in general.
Lines 324-327: Talking about “immunity” regarding the resistance of soft corals to acidification is an overstatement (for example see Inoue et al., 2013). Please clarify.
Line 330: “soft corals respond differently”
Line 334-342: This last paragraph could be complemented. It is written following what could involve a loss of hard corals to the ecosystem, with a focus on the structure and mucus release. It would gain interest if information referring to soft corals is added in comparison. For example, does soft coral not provide any function to the ecosystem? What is the trend of mucus release of soft corals to the reef environment (Bednarz et al., 2012)? What are other parameters that could be affected by a take-over of soft corals (e.g. biogeochemical cycles)?

Additional comments

General comments:
This study measured pulsation and growth rates of a tropical common soft coral species in response to glucose addition and elevated seawater temperatures. Pulsating soft coral species are representatives of an important group of coral reef invertebrates which are largely overlooked. As such, the work furthers our understanding of the ecophysiology of soft corals. The study documents a neutral effect of glucose addition at ambient temperature on the survival, growth and pulsation of the colonies. Under increased temperatures, the authors revealed a beneficial effect of glucose addition to the survival and pulsation measurements. Interestingly, they suggest the pulsation rates to be an early warning signal of ocean warming effect on Xeniidae. The paper is well written and the information documented is of interests for PeerJ readers. I would however suggest some changes or information to be further discussed before acceptance.
My main points target:
1. A statement regarding growth rates to be improved for better clarity.
2. Physiological parameters investigated here are survival, growth and pulsation. The discussion would gain even more interest if report of bleaching (or not) between the treatments could be added.
3. Addition of information on Xeniidae before extrapolating to soft corals in general and further discussing differences between hard and soft corals.

Reviewer 2 ·

Basic reporting

The manuscript addresses revenant and highly significant questions related to the effects of organic eutrophication on the resilience of a soft coral under ocean warming conditions. This topic is of significance as the existing evidence and predictions indicate shift in structure of benthic communities on coral reefs, manifested by decrease in hard coral abundance and a transition toward soft corals, algae and sponges. The manuscript experimentally tests the individual and the combined effects of both eutrophication and temp. rise and revealed some novel phenological response of a soft coral. The study lacks an overall hypothesis, yet addresses specific questions (Line 100-103). Surprisingly, the authors do not refer to studies showing resilience of soft corals to ocean acidification (e.g., https://www.nature.com/articles/nclimate1855) which are highly relevant to the topic. .

Experimental design

In general, the Material and Methods section is clear and it provides the required information. However, the experimental set up does not mention the geographic origin of the colonies and the environmental conditions under which they have been maintained since collection from the reef (line 110-120). These details might have affected the results of the experiments.
The authors do not explain why the pulsation was recorder at about noon time (line 169), has it been any reason for that? Similarly, why after day 7 the pulsation was only counted for 30 seconds (line 171-72). And more generally, how the time duration was determined?
This section, as well other sections, in the text erroneously mix up the terms "physiological data" (line 119) and "ecological assessments" (line 164). I couldn't find in this manuscript any real physiological results, unless the authors refer to pulsation as such. Yet, the authors did not show the physiological response of the soft coral. In short, the authors mix up phenological results with lack of physiological ones.





.

Validity of the findings

The presented results are meaningful and rather robust but as has already been stated above they are not always presented in the relevant context.

The authors should better explain how the have scored the damage to the colonies. (line 193-198) Mortality is an obvious stage, but for sublethal effects on the colonies I would expect further details.
The findings could have been much more convincing if physiological parameters were tested. Within the duration of the experiment, it can be assumed that symbiont (zooxanthellae) count and chlorophyll concentration would provide significant physiological criteria and probably substantiating the major findings and the emerged patterns. At present, the phenology of the tested soft coral supports the overall goal of the manuscript, but certainly it still lacks supportive evidence based on physiological or eco-physiological grounds (e.g., line 31). Similarly, the phenological results do not support lack of "disturbance of the soft coral holobiont" (line 233-234). Only after a thorough examination of the symbiotic algae (zooxanthella) such a conclusion can be presented. Saying that, the first paragraph of the discussion (line 288 onwards, especially lines 241-245), may provide supportive evidence in the current study for answering the first question (line 228-129).
The second question (line 266 onwards) is a critical one and the phenological results suggest that glucose addition supports Xenia to maintain growth and increase the ability to withstand the effect of elevated temp. (line 303-306). However, the authors ignore the critical and pressing issue that increased anthropogenic CO2 in the atmosphere results in a concurrent increase in sea surface temperatures (SST) and reduced seawater pH. Indeed, SST is predicted to rise by 1- 4 °C by the end of the 21st century by the continuous uptake of CO2 by the ocean’s surface alters the seawater carbonate system. Only, understanding the combined effects of the predicted end-of-the-century SST and pH conditions on a greater number of organisms, including soft corals, is therefore of great importance for predicting the fate of coral reefs. The manuscript does not refer to this point, but focuses on eutrophication and temperature.

Additional comments

In light of the above comments, it is suggested to alter the overall context of the manuscript and rewrite it in a context of phenological response of a soft coral to both studied parameters. Then, the significance of the study should reflect the results also using the appropriate and relevant terms. Not much is known on the effect of global change on soft corals compared to stony corals therefore the results presented in the current manuscript are significant. However, they can be considered for publication only when presented in a correct context.

---

## Round 0.2 · accepted · Accept

I have now heard back from the referees, and I concur with them that the revised manuscript is acceptable for publication. One referee has provided some feedback for minor corrections to the manuscript, but it seems to me that these a minor enough that they could be incorporated during production, so I am happy to notify you that the paper is accepted and move it forward in the process.

Reviewer 2 ·

Basic reporting

The authors responded to all my comments. Their rebuttal letter is comprehensive and refers to all the points.
However, I still have some comments as listed below:
1. The authors indicate that the tested colonies were originally collected in the Red Sea. This region extends along >15 latitudinal degrees and represents a variety of environmental parameters, including nutrient levels. Therefore, a more precise location where the colonies were obtained should be included in this manuscript: e.g., .Journal of Biogeography (2016) 43, 423–439; Mar Biodiv (2017) 47:991–993.,

2. Please note that "spp." shouldn't be in italic.
3. I think that the following two references are directly related to the study and should be incorporated into the text:
Y. Gabay, M. Fine, Z. Barkay, Y. Benayahu
Octocoral tissue provides protection from declining oceanic Ph
DOI: 10.1371/journal.pone.0091553
PLoS ONE, Vol. 9, 2014 (e91553)

Y. Gabay, Y. Benayahu, M. Fine
Does elevated pCO2 affect reef octocorals?
DOI: 10.1002/ece3.351
Ecology & Evolution, Vol. 3, 2012 (pp. 465-473)

Experimental design

See above

Validity of the findings

See above

Additional comments

See abov